# Proteomics Reveals mRNA Regulation and the Action of Annexins in Thyroid Cancer

**DOI:** 10.3390/ijms241914542

**Published:** 2023-09-26

**Authors:** Margarida Coelho, João Capela, Sandra I. Anjo, João Pacheco, Margarida Sá Fernandes, Isabel Amendoeira, John G. Jones, Luís Raposo, Bruno Manadas

**Affiliations:** 1CNC—Center for Neurosciences and Cell Biology, University of Coimbra, 3004-517 Coimbra, Portugal; margarida.serracoelho@gmail.com (M.C.);; 2CIBB—Centre for Innovative Biomedicine and Biotechnology, University of Coimbra, 3004-517 Coimbra, Portugal; 3III Institute for Interdisciplinary Research, University of Coimbra (IIIUC), 3030-789 Coimbra, Portugal; 4Department of Chemistry, Faculty of Sciences and Technology, University of Coimbra, 3004-535 Coimbra, Portugal; 5Centro Hospitalar Universitário São João, 4200-319 Porto, Portugal; 6I3S, Instituto de Investigação e Inovação em Saúde, 4200-135 Porto, Portugal; 7Ipatimup, Instituto de Patologia e Imunologia Molecular da Universidade do Porto, 4200-465 Porto, Portugal; 8Portuguese Society of Endocrinology, Diabetes and Metabolism, 1600-892 Lisbon, Portugal; 9EPIUnit-Institute of Public Health, University of Porto, 4050-600 Porto, Portugal

**Keywords:** proteomics, screening, thyroid cancer, annexins, Sp1

## Abstract

Differentiated thyroid cancer is the most common malignancy of the endocrine system. Although most thyroid nodules are benign, given the high incidence of thyroid nodules in the population, it is important to understand the differences between benign and malignant thyroid cancer and the molecular alterations associated with malignancy to improve detection and signal potential diagnostic, prognostic, and therapeutic targets. Proteomics analysis of benign and malignant human thyroid tissue largely revealed changes indicating modifications in RNA regulation, a common cancer characteristic. In addition, changes in the immune system and cell membrane/endocytic processes were also suggested to be involved. Annexin A1 was considered a potential malignancy biomarker and, similarly to other annexins, it was found to increase in the malignant group. Furthermore, a bioinformatics approach points to the transcription factor Sp1 as being potentially involved in most of the alterations seen in the malignant thyroid nodules.

## 1. Introduction

Thyroid cancer is the most common malignancy of the endocrine system. Papillary thyroid cancer (PTC) occurs in approximately 85–90% of all thyroid tumours [1,2,3] and is the most common thyroid carcinoma, followed by follicular thyroid cancer (FTC). Both these carcinomas are well-differentiated and typically have a better prognosis than undifferentiated cancers. They originate from follicular epithelial thyroid cells and are usually driven by BRAF- or RAS-events. However, as these genetic events take place, changes to the proteome are the first to become apparent with phenotypic changes..

The proteomics of thyroid cancer is relatively well-developed, with several reviews on this subject [4,5,6,7]. One of the earliest examples was able to distinguish differentially abundant proteins in follicular adenomas PTC, FTC, and non-tumour thyroid tissues, differentiating in particular FTC from follicular adenomas and FTC from PTC [8]. Another study that reported proteomic profiling of four types of thyroid cancer: papillary, follicular, anaplastic, and medullary—as well as benign thyroid lesions—revealed that medullary and anaplastic carcinomas featured proteins associated with neuroendocrine functions. Follicular and papillary carcinomas had similar proteomic profiles and factors typically associated with advanced malignancies, while that of follicular adenomas was highly similar to follicular carcinoma [9]. Martinez-Aguilar et al. were able to quantify over 1600 proteins and associated PTC with the disruption of cell contacts (loss of E-cadherin), actin cytoskeleton dynamics and the loss of differentiation markers [10]. Although interesting for the characterization of each type of thyroid carcinoma, these studies did not focus on distinguishing benign tumours (including not only adenomas but also goitre) from the main malignant lesions of the thyroid (differentiated thyroid carcinomas, including PTC and FTC). Proteomics has been reported to provide diagnostic and prognostic biomarkers and reveal potential therapeutic targets that may only be discovered when collectively studying malignancy adaptations [11].

In this work, liquid chromatography coupled to tandem mass spectrometry (LC-MS/MS) using a label-free bottom-up approach that combines data-dependent acquisition (DDA) with data-independent acquisition (DIA, sequential window acquisition of all theoretical fragment-ion spectra–SWATH) was used. The purpose of this study was to determine the proteome differences between benign and malignant tissue lesions of the thyroid to better understand which processes could collectively be affected in thyroid cancer.

## 2. Results

Samples from individuals undertaking thyroid surgery were collected and assigned to the benign group if histological diagnosis revealed the sample was an adenoma or thyroid follicular nodular disease, or to the malignant group if the samples were considered a differentiated thyroid carcinoma, either papillary or follicular (Table 1 and Appendix A). A proteomics analysis was then performed on these thyroid tissue samples that included either benign or malignant lesions. A total of 3405 proteins were identified, of which 2423 were quantified after the peptide quality threshold.

After a Mann–Whitney test, 1629 proteins were considered altered between the two groups. Gene ontologies altered by the malignancy process were categorized into molecular function and biological process (Figure 1). The percentages of gene hits against the total number of molecular function and biological process hits revealed that generally the altered proteins have binding and catalytic activity and are involved in cellular and metabolic processes. In addition to this analysis, a FunRich analysis signalled that 17.4% of the altered proteins were related to protein metabolism, 16.6% to intermediary metabolism, 16.4% to energy pathways, and 9% to cell growth and/or the maintenance as significant biological processes (Appendix A).

A ReactomeGSA analysis revealed the overall regulation of several pathways using values for differential expression (Figure 2). The immune system was signalled as being highly affected, particularly the innate immune system (Appendix A), which is the first line of defence against malignancy and is also critical for activating an adaptive immune response. However, while the activation of adaptive immune cells may eradicate malignant cells, the chronic activation of various innate immune cells can promote tumour development [12].

In order to understand the systemic biological significance behind altered proteins in malignant versus benign lesions, different software tools that link the proteins of interest to biological pathways or processes were used since they can provide different perspectives on these data. The GOrilla web-based application revealed that the biological processes which change the most due to tissue malignancy are related to RNA regulation (Appendix A and Appendix A). Although mRNA processing is fundamental for gene expression, mutations in RNA splicing factor genes or the shortening of 3′ untranslated regions have been observed in the cancer phenotype [13]. Viral processes were also found to be perturbed, which is likely related to endocytosis mechanisms common to viral infection, which have been dysregulated in cancer, promoting the migration and invasion of cancer cells [13]. Specific KEGG pathways were enriched in malignancy (Appendix A) when using a DAVID tool. The ribosome proteins were the most featured, followed by spliceosome (Figure 3 and Appendix A). Alterations to the ribosome likely lead to a deregulation of the p53 tumour suppressor protein resulting in altered mRNA translation and dysregulated protein synthesis [14,15]. On the other hand, alterations to the spliceosome can also affect translation through changes in pre-mRNA splicing. Although not significantly enriched, “Pathways in cancer” revealed alterations to proteins associated with proliferation, tissue invasion and metastases, evading apoptosis, sustained angiogenesis, genomic instability, insensitivity to anti-growth signals, and the block of differentiation (Appendix A). These alterations in biological processes concerning metabolism and cell regulation are consistent with malignancy processes discussed previously, since a metabolic adaptation to abnormal cell growth and proliferation is a hallmark of cancer.

A protein interaction network was constructed with only the proteins that best distinguished benign from malignancy samples (significantly altered proteins with PLS-DA VIP > 1 (1st component) and at least 100% fold change) (Figure 4a). Three main observable clusters reflect the following molecular function gene ontologies: RNA binding, protein-containing complex binding, and catalytic activity. Collectively, the proteins found to be statistically different between benign and malignant tissues suggest the activity of transcription factor Sp1 (53.8%) in most of these results (Figure 4b). This transcription factor has been previously associated with poor prognosis in cancer, but because it can either activate or inhibit the expression of several essential oncogenes and tumour suppressors, a more complete understanding is necessary before it can be used as a therapeutic target [16].

Receiver operating characteristic (ROC) curves for single biomarkers were plotted for the 1629 statistically different proteins. Annexin A1 produced the best ROC with 0.993 area under the ROC curve (AUC) (95% confidence interval: 0.968–1) (Figure 5a and Appendix A). This specific upregulation in carcinomas of follicular cell origin has been reported previously [17,18,19] and suggests that annexin A1 could be a prognostic biomarker for thyroid cancer. This calcium and phospholipid-binding protein participates in inflammatory processes, cell proliferation modulation, cell death regulation, and tumour formation and development. Besides annexin A1, other annexins (annexins A2, A4, A5, A6, A7, A9, and A11) were upregulated in the malignant group, with the exception of A3 which was not considered statistically significant (Figure 5b). These calcium-dependent phospholipid-binding proteins which interact with various cell-membrane components involved in the cell’s structural organization, enzyme modulation, and ion fluxes can alter intracellular signalling. Several members of this family have been linked to cancer, with annexin A2 being one of the most studied annexins and a known cancer biomarker [20,21,22], and annexin A4 also being associated with different cancers such as colorectal, gastric [23], and thyroid cancer [24]. Annexin A4 is reported to be highly expressed in follicular and medullary carcinoma but not in other thyroid carcinoma subtypes—precisely the opposite of our results. Annexin A5 is generally upregulated in a variety of cancers, but in the case of thyroid carcinoma, it has been negatively correlated with malignancy, with lower expression in FTC than PTC or follicular adenomas [8]. Annexin A6 has dual functions, acting either as a tumour promoter or tumour suppressor depending on the type of cancer or malignancy [25]. Similarly, annexin A7 appears to have a suppressor or promotor role depending on the type of cancer [26]. Annexin A9 has not been as extensively studied, but it has been reported as a predictor of prognosis in colorectal cancer [27] and epithelial cancer [28]. Annexin A11 plays an essential role in cytokinesis; without it, cell division cannot be concluded, leading to apoptosis. In cancer, its upregulation promotes cell proliferation, while its suppression is associated with reduced cell proliferation. Thus, annexin A11 serves as a diagnostic biomarker and may also represent a therapeutic target [29,30].

## 3. Discussion

Proteomics approaches have been previously applied in the study of thyroid cancer tissues [4,5,6,7]. Advancements in equipment sensitivity as well as developing the combination of dependent and independent acquisitions (combining DDA for identification and DIA/SWATH for quantification) led to the quantification of a total of 2423 proteins. The most similar publication to our work is that of Martinez-Aguilar et al., which also used SWATH mass spectrometry, where 1629 proteins were quantified in a total of 32 specimens, equally distributed between normal histology, follicular adenoma, FTC, and classic PTC (n = 8 per group) [10]. However, in the present work we were able to quantify more proteins confidently and focused on the comparison between benign and malignant (FTC and PTC) because this has the greatest clinical interest in terms of diagnostic value.

The main biological processes that were affected by malignancy were related to RNA regulation, particularly through changes in splicing-associated proteins, endocytosis and the immune system. Removing introns from messenger RNA precursors is essential for regulating protein expression, so splicing perturbations contribute to cancer development [31]. On the other hand, the lack of nutrients in the tumour microenvironment forces the degradation of macromolecules for resupply, for example, intracellular proteins can be recycled through autophagy and degraded by lysosomes to supply amino acids to the cancer cells, which corroborates with the increase in endocytosis and other viral pathways [32]. Additionally, other changes detected in the malignant group also cause the immune system to respond [33].

Sp1, a prominent transcription factor, stood out as one of the main transcription factors involved in the regulation of the proteins found altered in this study through a bioinformatic analysis of the affected proteins. It belongs to the Sp-family of transcription factors, which includes Sp2, Sp3, and Sp4. This family plays pivotal roles in cell cycle control and developmental processes [34]. However, Sp1 is often found to be overexpressed in cancer, making it a negative prognostic factor [16]. In the context of thyroid cancer, Sp1 displays distinctive expression patterns [35,36] and has been identified as a participant in a positive feedback loop that regulates malignant characteristics [37]. The target genes regulated by Sp1 predominantly relate to cell proliferation and oncogenesis, which correspond to cancer’s hallmarks. Nevertheless, these genes also participate in crucial cellular processes such as proliferation, differentiation, the response to DNA damage, apoptosis, senescence, and angiogenesis. This complex involvement makes it difficult to target Sp1 as a therapeutic target.

Annexin A1, a protein that presented the best ROC AUC of this dataset, is a member of a family of Ca^2+^-regulated phospholipid-dependent and membrane-binding annexin proteins. Annexins can have several functions, such as a membrane scaffold to induce changes in the cell’s shape, trafficking and organization of vesicles by exo- and endocytosis, and homeostatic regulation of intracellular Ca^2+^ concentration. Alterations to the expression of annexin A1 have been associated with malignancy, and particularly with disease severity. Annexin A1’s role in cancer includes regulation of cellular proliferation, metastasis, lymphatic invasion, development of resistance to anti-cancer treatment, and modulation of cancer-related signalling pathways [38]. In thyroid cancer, this protein has been reported as a potential diagnostic and prognostic biomarker of PTC by regulating epithelial-mesenchymal transition and activating the IL-6/JAK2/STAT3 pathway [19].

One disadvantage of this study is the low sample size, particularly for the malignant group with 11 samples. The benign group did not include all harvested samples because samples from thyroids containing a malignant lesion were excluded due to possible contaminations, thus leaving 60 samples. All extracted nodules, including more than one nodule per individual, were included. Because of the metabolic heterogeneity of cancer, each nodule may have its own proteomic/metabolomic phenotype not only between nodules, but even within the same nodule [39], and on this basis, it is justified that each nodule represents an independent sample. Another disadvantage of the study is that most of the malignant group samples were PTC lesions, with only one FTC. Given the low number of samples, PTC and FTC samples were grouped as differentiated thyroid carcinomas. Had FTC not been a minority in the malignant group, some proteins might not have been differently expressed between benign and malignant groups. The lack of FTC specimens is possibly related to the demographics of the population studied since most samples come from participants being followed at a medical facility in the city of Porto in Portugal, where these cases are typically considered rare [40].

To our knowledge, this study is the most comprehensive proteomic screening to compare benign and malignant thyroid lesions. A gene ontology and pathway analysis of statistically different proteins according to malignancy mostly revealed changes that suggest alterations in mRNA translation, a typical cancer hallmark. Annexin A1 was also found to be elevated in malignancy, along with other annexins, suggesting changes to the cell membranes and cytoskeleton. Not only can this and other proteins be considered potential biomarkers of malignancy, but they can also be potential therapeutic targets, as also demonstrated by the transcription factor Sp1. This work has therefore contributed to an overview of dysregulated pathways and knowledge on possible therapeutic targets of differentiated thyroid cancer.

## 4. Materials and Methods

Patients: The study was approved by the Ethics Committee of the Centro Hospitalar Universitário de São João/Faculdade de Medicina da Universidade do Porto (approval ID 125/18). Informed consent was obtained from all participants. The cohort of thyroid tissue lesions consisted of 71 nodules from 43 patients. The benign group consisted of follicular adenoma and follicular nodular disease, while the malignant group comprised differentiated carcinomas of follicular cells (papillary carcinoma and follicular carcinoma). In some cases, more than one nodule per individual was studied. Groups were gender- and age-matched (Table 1). Tissue samples were obtained during surgical resection and immediately stored at −80 °C until analysis. The final diagnosis was obtained after a postoperative histopathological examination of the same lesion (Appendix A).

Tissue preparation: Briefly, after non-destructive high-resolution magic angle spinning (HR-MAS) ^1^H nuclear magnetic resonance (NMR) analysis, nodules were recovered from the rotor and stored in 100 μL of 0.5 M triethylammonium bicarbonate (TEAB) solution with protease inhibitors (cOmplete™, ethylenediamine tetraacetic acid (EDTA)-free Protease Inhibitor Cocktail, Roche). Samples not analysed using HR-MAS NMR were also added to the same solution. Tissue samples were homogenised with the Dispersing-aggregates POLYTRON^®^ PT1200 E with a 3 mm tip (Kinematica AG, Malters, Switzerland). The homogenised mixture was then centrifuged at 5000× *g* for 5 min at 4 °C. The supernatant was harvested, with 5 μL being used for total protein content assessment with the Pierce™ 660 nm Protein Assay Reagent (ThermoFisher™, Waltham, MA, USA), according to the manufacturer’s instructions. A volume corresponding to approximately 100 μg of protein was harvested from each sample to continue sample processing. Moreover, pools of benign and malignant lesions were created using 5 μL from selected samples. To each individual and pooled sample, 2 μg of the recombinant protein green fluorescent protein and maltose-binding periplasmic protein (MBP-GFP) were added as an internal standard [41]. Protein precipitation was performed using 400 μL of cold methanol. Samples were incubated overnight at −80 °C and then centrifuged at 20,000× *g* for 20 min at 4 °C.

Proteomics by LC-MS/MS: The pellet fraction, containing the proteins, was resuspended in 30 μL of 2× Laemmli sample buffer via sonication with a Vibra cell 75041 cup horn (Bioblock Scientific, Illkirch, France). Samples were incubated at 95 °C for 5 min in a Thermomixer comfort (Eppendorf, Hamburg, Germany) and 2 μL of 40% acrylamide (Bio-Rad Laboratories, Lda., Hercules, CA, USA) was added as an alkylating agent. Individual and pooled samples were loaded into an SDS-PAGE 4–20% Mini-PROTEAN^®^ TGX™ precast gel and run at 110 V [42] (Appendix A). Gel staining was performed as previously described [43]. Each lane was divided into fractions, and each fraction was divided into smaller pieces with the help of a scalpel and added to a 96 multi-well plate containing ddH_2_O. After destaining the gel pieces with 50 mM ammonium bicarbonate and 30% acetonitrile, in-gel digestion and peptide extraction were performed as previously described [43]. Peptides were evaporated in the Concentrator Plus/Vacufuge^®^ (Eppendorf, Hamburg, Germany), and resolubilized in 30 μL of 2% acetonitrile and 0.1% formic acid aided via sonication with a Vibra cell 75041 cup horn (Bioblock Scientific, Illkirch, France) at 20% amplitude pulse every 1 s for 2 min. After centrifugation at 14,100× *g* for 5 min at room temperature, samples were transferred to vials for LC-MS analysis. Samples were analysed on a NanoLC™ 425 System (Eksigent^®^, Framingham, MA, USA) coupled to a TripleTOF™ 6600 System (Sciex^®^, Framingham, MA, USA) using DDA for each fraction of the pooled samples for protein identification and DIA/SWATH-MS acquisition of each individual sample for protein quantification. Detailed procedures of data acquisition are described in Section A.1. Peptide identification and library generation were performed using ProteinPilot™ 5.0 software (Sciex^®^, Framingham, MA, USA), while data processing for quantification was performed using the SWATH™ processing plug-in for PeakView™ 2.2 (ABSciex^®^, Framingham, MA, USA). Detailed procedures of data processing are described in Section A.2. The mass spectrometry proteomics data have been deposited in the ProteomeXchange Consortium via the PRIDE [44] partner repository with the dataset identifier PXD035583.

Data analysis: Multivariate analysis was performed in Metaboanalyst 5.0 [45]. Log transformation and Pareto scale were performed for partial least squares discriminant analysis (PLS-DA). Receiver operating characteristic (ROC) analysis was also performed in this platform, with the same normalization parameters.

Due to the small number of samples in each group and the lack of normal distribution of the populations, a Mann–Whitney test (univariate analysis) was applied to test differences between groups using IBM SPSS Statistics 23 (IBM^®^, Armonk, NY, USA). Violin plots of relative abundance values were obtained using GraphPad Prism 6.0 (GraphPad Software Inc., San Diego, CA, USA).

The Database for Annotation, Visualization and Integrated Discovery (DAVID) 6.8 online tool assisted with the evaluation of biological processes, gene ontology (GO) terms, and Kyoto Encyclopedia of Genes and Genomes (KEGG) pathways using a list of statistically different proteins. For analysis of the enriched GO terms, the web-based application Gene Ontology enRIchment anaLysis and visuaLizAtion tool (GOrilla) [46], the Functional Enrichment analysis tool (FunRich) 3.1.3 [47], and Protein ANalysis THrough Evolutionary Relationships (PANTHER) 17.0 [48] were used. Association of transcription factors was performed on FunRich.

The STRING 11.5 web-based application [49] was used to create the protein interaction networks with enriched ontology terms. The network was constructed using significantly altered proteins with PLS-DA VIP > 1 (1st component) and at least 2-fold change. An interaction score of 0.7 was used and disconnected nodes were not shown.

Open-source Reactome 83 was used to visualize pathways coverage and enrich those pathways with colour coding relative to fold-change. For differential expression analysis, the ReactomeGSA was used [50]. All quantified proteins were submitted, non-normalized to total intensity, transformed by log_2_ and grouped by benign versus malignant. Normalization was performed using a continuous scale built-in function of the online tool.

Kyoto Encyclopedia of Genes and Genomes (KEGG) Mapper–Search&Color Pathway tool 5 was used in the analysis of functional pathways.

## Figures and Tables

**Figure 1 ijms-24-14542-f001:**
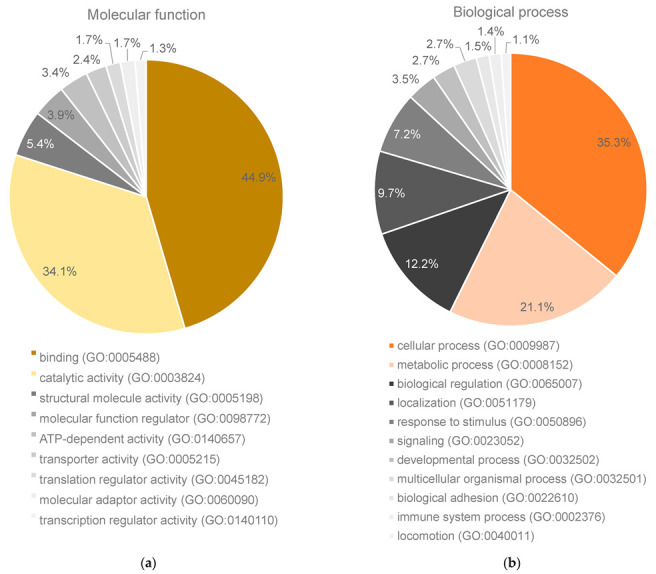
Percentage of gene hits for the proteins altered in the malignant group show (**a**) binding and catalytic activity as the main molecular functions and (**b**) cellular and metabolic processes as the main biological processes. Representation of PANTHER results with at least 1% hits in a pie chart.

**Figure 2 ijms-24-14542-f002:**
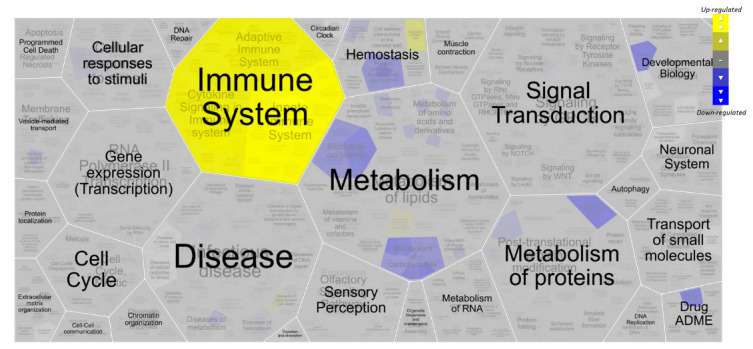
Immune system is altered in malignant thyroid lesions. Voronoi diagram with hierarchical view of Reactome’s differentially regulated pathways due to thyroid tissue malignancy.

**Figure 3 ijms-24-14542-f003:**
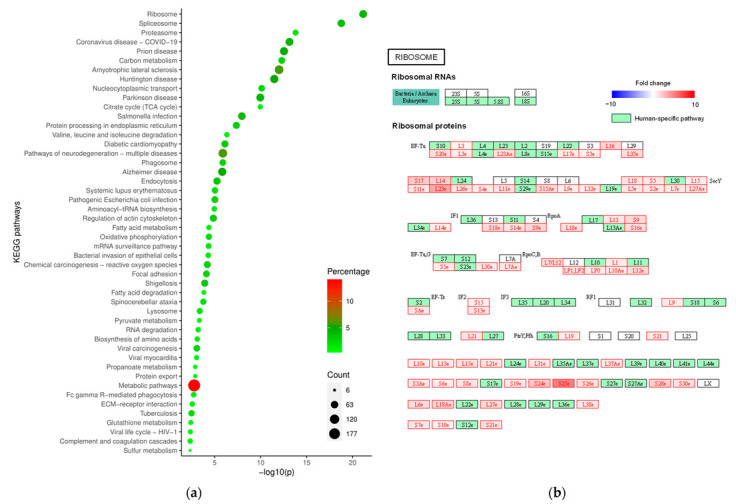
Enriched KEGG pathways in malignant thyroid lesions show the ribosome pathway as the most significantly affected. (**a**) Representation of DAVID functional annotation of KEGG pathways. Count (circle size) corresponds to the number of proteins in the submitted list that belong to each pathway; percentage (colour) is the number of proteins in each pathway relative to the total of imported proteins and −log10(p) (axis) is the results of Fisher’s exact test to determine gene enrichment; (**b**) ribosome KEGG pathway with annotation of statistically altered proteins in malignant thyroid lesions. Proteins quantified in this study are labelled as red if they are increased in the malignancy group and blue if they are decreased, with this colour code gradient depending on fold change value. Other proteins from human-specific pathways but not quantified in this study are depicted in green.

**Figure 4 ijms-24-14542-f004:**
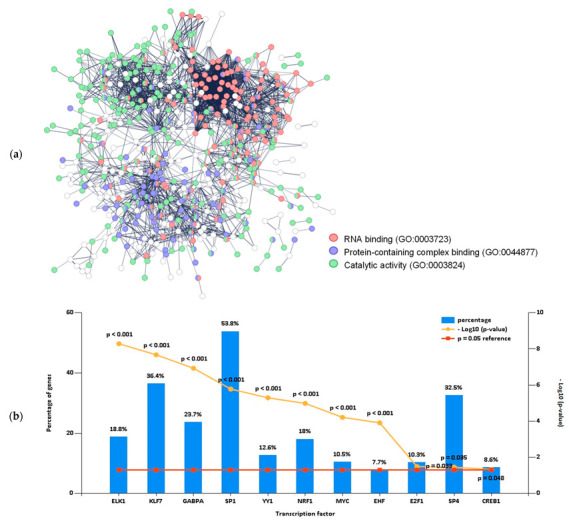
Clusters of proteins related to RNA binding, protein-containing complex binding, and catalytic activity and potential transcription factors were found in malignant thyroid lesions. (**a**) Protein interaction network, where each protein is coloured with the gene ontology of the molecular function that is predominant in each cluster: RNA binding (red), protein-containing complex binding (blue), and catalytic activity (green); (**b**) selection of potential transcription factors altered by malignancy with a *p* < 0.05 for the enrichment performed with a hypergeometric test (HG).

**Figure 5 ijms-24-14542-f005:**
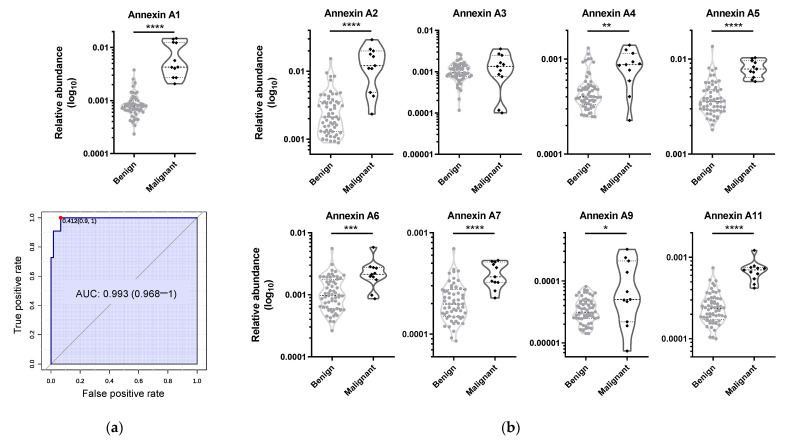
Annexin A1 presented the best univariate ROC AUC to distinguish malignant thyroid lesions from benign. (**a**) Violin plot of annexin A1 and the respective ROC curve; (**b**) violin plots of annexins A2, A3, A4, A5, A6, A7, A9, and A11. * *p* ≤ 0.05, ** *p* ≤ 0.01, *** *p* ≤ 0.001, **** *p* ≤ 0.0001 (Mann–Whitney test).

**Table 1 ijms-24-14542-t001:** Demographic summary of the sample cohort.

Group	Sex	Female (%)	Age (Years)	BMI (kg/m^2^)	Free T4 (ng/dL)	TSH (mU/L)
Benign	Female = 24 Male = 9	72.7	58.4 ± 2.3	28.7 ± 0.9 **	1.11 ± 0.07	0.83 ± 0.12
Malignant	Female = 8 Male = 2	80.0	53.2 ± 7.1	23.7 ± 1.0	1.08 ± 0.04	1.28 ± 0.29

Note: data represents mean ± standard error of mean. ** *p* ≤ 0.01 (Mann–Whitney test group comparison used for age, BMI, free T4, and TSH). Abbreviations: body mass index (BMI), thyroid-stimulating hormone (TSH), and thyroxine (T4).

## Data Availability

The data presented in this study are openly available in the ProteomeXchange Consortium via the PRIDE partner repository with the dataset identifier PXD035583.

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
