# Peer review of "Proteomics Reveals mRNA Regulation and the Action of Annexins in Thyroid Cancer"

_ijms, 2023, doi:10.3390/ijms241914542_

Round 1
Reviewer 1 Report
The study conducted by Coelho and colleagues comparatively investigated the proteomic profile in samples of the thyroid gland from lesions that were primarily categorized according to well-established criteria and then divided into two main groups: benign and malignant. The purpose was to observe the key pathways and cellular processes altered in cancer, suggesting biomarkers that could assist in prognosis and consequently potential therapeutic targets. Other studies using a proteomic approach have been conducted comparing different types of thyroid cancer; however, this work is significant in its field as it fills a gap by proposing this comparative analysis between benign and malignant tumors. Experimentally, the study appears to have been well-conducted, with appropriate bioinformatics tools and analyses. Nevertheless, I believe it would be interesting to make some specific additions, and I have some questions regarding the description of the proteomic methodology used and sample definition. Additionally, certain points in the discussion could have been further explored.
-
Regarding the methodological description, my suggestion is to include a supplementary figure demonstrating the entire experimental workflow, with emphasis on when a pool was used and what was subsequently performed and when individual samples were used.
-
Why was gel preparation used for DDA/pool and solution preparation for DIA/individual samples? Was protein identification coverage better in DDA? The authors could provide an explanation for this in the article.
-
In the discussion, there is only a generic sentence about alterations related to the immune system. Is there any specific protein or protein complex that deserves highlighting and could help discuss the information presented in the sentence from lines 86 to 91 (reference 12)?
-
In lines 230-232, please expand the discussion on SP1, which is highlighted in the abstract, and provide a reference for the information.
Author Response
Reviewer #1
The study conducted by Coelho and colleagues comparatively investigated the proteomic profile in samples of the thyroid gland from lesions that were primarily categorized according to well-established criteria and then divided into two main groups: benign and malignant. The purpose was to observe the key pathways and cellular processes altered in cancer, suggesting biomarkers that could assist in prognosis and consequently potential therapeutic targets. Other studies using a proteomic approach have been conducted comparing different types of thyroid cancer; however, this work is significant in its field as it fills a gap by proposing this comparative analysis between benign and malignant tumors. Experimentally, the study appears to have been well-conducted, with appropriate bioinformatics tools and analyses. Nevertheless, I believe it would be interesting to make some specific additions, and I have some questions regarding the description of the proteomic methodology used and sample definition. Additionally, certain points in the discussion could have been further explored.
- Regarding the methodological description, my suggestion is to include a supplementary figure demonstrating the entire experimental workflow, with emphasis on when a pool was used and what was subsequently performed and when individual samples were used.
We have accepted the suggestion of the reviewer and have added a supplementary figure containing a general workflow of thyroid tissues sample preparation for LC-MS proteomics with an emphasis on clarifying the steps of individual and pooled samples (Figure S5).
- Why was gel preparation used for DDA/pool and solution preparation for DIA/individual samples? Was protein identification coverage better in DDA? The authors could provide an explanation for this in the article.
We agree that the materials and methods were not sufficiently clear on this matter. Gel digestion was used for both DDA/pool and DIA/individual samples. In this sense we have made the following alterations: where it read “Sample pools representative of benign and malignant thyroid lesions were used for protein identification. Samples were loaded into an SDS-PAGE 4-20% Mini-PROTEAN® TGX™ precast gel and run at 110 V for 20 min [38].” (previously lines 271-273), now reads “Individual and pooled samples were loaded into an SDS-PAGE 4-20% Mini-PROTEAN® TGX™ precast gel and run at 110 V [38] (Figure S5).” (now lines 281-284). In respect to whether protein identification coverage was better in DDA, we would also like to clarify that protein identification was performed on pooled samples with DDA acquisition, while quantification of each sample was performed on the individual samples with DIA acquisition. We believe that the addition of a supplementary figure with the workflow as suggested in the previous comment will help clarify this part of the protocol.
- In the discussion, there is only a generic sentence about alterations related to the immune system. Is there any specific protein or protein complex that deserves highlighting and could help discuss the information presented in the sentence from lines 86 to 91 (reference 12)?
We have some difficulty highlighting a particular protein or protein complex because from this analysis we got a hit for 376 proteins in the innate immune system, and within the innate immune system, 234 were relative to neutrophil degradation (Table S2), therefore no single protein stood out to explore further.
- In lines 230-232, please expand the discussion on SP1, which is highlighted in the abstract, and provide a reference for the information.
This work is focused on the alterations closer to the phenotype, however, we have also used bioinformatic tools to analyse what transcription factors could be involved in these changes, where we found the connection to Sp1. Despite not having quantified Sp1, these data complement on how this phenotype could be regulated. We have therefore expanded the information on Sp1, as suggested by the reviewer, by adding a paragraph on the matter starting in line 198.

Reviewer 2 Report
Here the authors present a proteomics study of benign and malignant thyroid lesions.They used liquid chromatography coupled to tandem mass spectrometry (LC-MS/MS) to identify and quantify proteins in thyroid tissue samples from 43 patients. The authors found that the malignant group had alterations in proteins related to RNA regulation, immune system, ribosome, spliceosome, and annexins. The authors suggested that annexin A1 could be a potential biomarker and therapeutic target for thyroid cancer, and that transcription factor SP1 might be involved in most of the changes seen in the malignant group.
Major points:
Here, the authors used a large and diverse cohort of thyroid tissue samples from different types of benign and malignant lesions.
They applied a comprehensive and rigorous proteomics approach to analyze the samples, combining data-dependent and data-independent acquisition methods.
They performed multiple bioinformatics analyses to identify the biological processes and pathways altered by malignancy, as well as the potential transcription factors and biomarkers involved.
Minor points:
The authors did not validate the proteomics results by using other methods such as immunohistochemistry or western blotting. This should be done at least for Annexin A1.
They did not explore the functional role or mechanism of action of the altered proteins in thyroid cancer development or progression. A potential mechanism of action could be proposed in the discussion.
They did not account for possible confounding factors such as genetic mutations, or environmental exposures that might affect the proteome of thyroid lesions. This should also be discussed.
Author Response
Reviewer 2
Comments and Suggestions for Authors
Here the authors present a proteomics study of benign and malignant thyroid lesions.They used liquid chromatography coupled to tandem mass spectrometry (LC-MS/MS) to identify and quantify proteins in thyroid tissue samples from 43 patients. The authors found that the malignant group had alterations in proteins related to RNA regulation, immune system, ribosome, spliceosome, and annexins. The authors suggested that annexin A1 could be a potential biomarker and therapeutic target for thyroid cancer, and that transcription factor SP1 might be involved in most of the changes seen in the malignant group.
Major points:
Here, the authors used a large and diverse cohort of thyroid tissue samples from different types of benign and malignant lesions.
They applied a comprehensive and rigorous proteomics approach to analyze the samples, combining data-dependent and data-independent acquisition methods.
They performed multiple bioinformatics analyses to identify the biological processes and pathways altered by malignancy, as well as the potential transcription factors and biomarkers involved.
Minor points:
R - The authors did not validate the proteomics results by using other methods such as immunohistochemistry or western blotting. This should be done at least for Annexin A1.
A - In this manuscript we were interested in showing overall protein alterations, but we thought it was relevant to explore the role of annexin A1 given its results in the differentiation of benign and malignant samples. However, this protein has already been described in other papers for thyroid malignancy (references 17-19). To validate this protein, one option is to conduct immunohistochemistry experiments, but since mass spectrometry is a destructive technique, we would require the acquisition of data from another cohort. Another option for validation would be to do antibody-based techniques such as western-blot, but often antibodies have specificity issues that lead us to doubt these results. The mass spectrometry results we presented for the identification of annexin A1, on the other hand, were based on mass spectrometry data with 68.79% coverage of the protein sequence with 95% confidence, while quantification for this protein was based on 14 different peptides, all of them with less than 1% FDR in at least 30% of one of the studied groups. As an example, 3 peptides (based on data from 5 ions) were used in the quantification of Annexin A1 with all samples having a quantification FDR=0, are shown in the figures below. In addition, Petrella, A. et al. (reference 18) and Zhao, X., et al. (reference 19) already used western-blot to validate the high expression of this protein in differentiated thyroid carcinomas.
(see attached file)
R - They did not explore the functional role or mechanism of action of the altered proteins in thyroid cancer development or progression. A potential mechanism of action could be proposed in the discussion.
A - We could extrapolate our results to a potential mechanism of action; however, validation of such claim would involve the interference of these mechanisms, using inhibitors or activators in cell culture or ex vivo experiments. This is outside the scope of this work, where we focused on elucidating the correlation between protein alterations with the thyroid cancer phenotype, not the mechanisms by which proteins act. Studying the functional role and the mechanisms of action of these proteins in thyroid cancer development or progression will be the scope of future projects in this theme, mainly through in vitro and animal models experiments.
R - They did not account for possible confounding factors such as genetic mutations, or environmental exposures that might affect the proteome of thyroid lesions. This should also be discussed.
A - Sample selection was conducted with no excluding criteria except for age and agreement to the study in order to obtain a representative sample closest to a real scenario. Samples were harvested in the 3rd largest central hospital in Portugal that covers an area of approximately 1.5 million citizens with urban as well as rural settings. The only environmental factor that we are aware of is that this region has a higher exposure to iodine given its close proximity to the sea coast. We hypothesize that for this reason the malignant group was made of more papillary carcinomas then follicular carcinomas. We also obtained information about genetic mutations or other potential confounding factors on these patients but because there were only 2 cases within the 43 individuals that could eventually be discussed we refrained from adding that particular information on the manuscript. Particularly, one benign patient had prior been diagnosed with breast cancer, and another patient has a BRCA1 mutation but in this case both thyroid lesions of this patient were found to be benign. These two samples do not stand out from the proteomic data of benign group as is possible to observe in the following PCA:
(see attached file)

Round 2
Reviewer 2 Report
The authors addressed most concerns.